# Fabrication of X-ray Gratings for Interferometric Imaging by Conformal Seedless Gold Electroplating

**DOI:** 10.3390/mi12050517

**Published:** 2021-05-07

**Authors:** Konstantins Jefimovs, Joan Vila-Comamala, Carolina Arboleda, Zhentian Wang, Lucia Romano, Zhitian Shi, Matias Kagias, Marco Stampanoni

**Affiliations:** 1Paul Scherrer Institut, 5232 Villigen, Switzerland; joan.vila-comamala@psi.ch (J.V.-C.); caroarboleda@gmail.com (C.A.); zhentian.wang@psi.ch (Z.W.); lucia.romano@psi.ch (L.R.); zhitian.shi@psi.ch (Z.S.); matias.kagias@psi.ch (M.K.); marco.stampanoni@psi.ch (M.S.); 2Institute for Biomedical Engineering, University and ETH Zürich, 8092 Zürich, Switzerland; 3Department of Physics and CNR-IMM, University of Catania, 64 via S. Sofia, 95123 Catania, Italy

**Keywords:** X-ray interferometry, phase contrast imaging, grating, high aspect ratio, deep reactive ion etching, Bosch process, silicon, gold, seedless electroplating, visibility

## Abstract

We present a method to produce small pitch gratings for X-ray interferometric imaging applications, allowing the phase sensitivity to be increased and/or the length of the laboratory setup to be minimized. The method is based on fabrication of high aspect ratio silicon microstructures using deep reactive ion etching (Bosch technique) of dense grating arrays and followed by conformal electroplating of Au. We demonstrated that low resistivity Si substrates (<0.01 Ohm·cm) enable the metal seeding layer deposition step to be avoided, which is normally required to initiate the electroplating process. Etching conditions were optimized to realize Si recess structures with a slight bottom tapering, which ensured the void-free Au filling of the trenches. Vapor HF was used to remove the native oxide layer from the Si grating surface prior to electroplating in the cyanide-based Au electrolyte. Fabrication of Au gratings with pitch in the range 1.2–3.0 µm was successfully realized. A substantial improved aspect ratio of 45:1 for a pitch size of 1.2 µm was achieved with respect to the prior art on 4-inch wafer-based technology. The fabricated Au gratings were tested with X-ray interferometers in Talbot–Laue configuration with measured visibility of 13% at an X-ray design energy of 26 keV.

## 1. Introduction

X-ray imaging is a powerful non-destructive method, capable of providing information on the inner structure of specimens. It is an invaluable tool for medical diagnostics, material science, and security. The contrast mechanism in X-ray images is usually based on X-ray absorption. However, when the object consists of a weakly absorbing matter with small variation in optical properties (such as carbon reinforced polymers [1] or biological soft tissues [2]), the X-ray absorption contrast mechanism becomes ineffective. X-ray grating-based interferometric imaging can overcome these limitations by providing additional information about the phase of the X-ray wave propagating through the object, which can increase the contrast between materials with similar absorption levels. The method was first demonstrated on synchrotrons [3] and later translated to laboratory sources [4,5]. Since the initial demonstrations, grating based imaging has shown a great potential in medical diagnostics [6,7,8,9]. However, the reliable fabrication of gratings remains the bottleneck for a massive exploitation and a final commercialization of X-ray grating interferometry (GI) based systems. X-ray GI benefits from decreasing the grating pitch: on the one hand, the smaller the grating pitch the better the interferometer’s phase sensitivity by detecting smaller refraction angles produced by the sample; on the other hand, the smaller the grating pitch the shorter the X-ray interferometer, which is especially convenient to overcome the limitations of low flux delivered by laboratory X-ray sources. Last, but not least, shorter interferometers have a direct impact on the footprint of the system, which makes it more attractive from a practical point of view. However, the fabrication of gratings with pitches below 2 µm remains a challenge. In the following, we describe the state of the art of the small pitch gratings fabrication and the limitations of specific technologies.

Deep X-ray lithography in combination with gold electroplating, also known as X-ray LIGA from the German acronym Lithographie, Galvanoformung, Abformung (Lithography, Electroforming, Molding) is a benchmark for production of X-ray gratings with a pitch in the range of a few micrometers and above [10]. However, LIGA requires synchrotron radiation for the X-ray lithography step and the achievable aspect ratio as well as the reliability of the process rapidly drops at pitch values below about 2 µm due to a limited mechanical stability of the polymer structures with a maximum reported aspect ratio of 10:1 for a pitch size of 1.2 µm [11].

Another challenge in grating fabrication is the overcoating or filling of the weakly X-ray absorbing silicon structures with highly absorbing metals. Microstructures in monocrystalline silicon are usually more stable than polymers, so higher aspect ratios and smaller pitches are achievable in silicon structures in comparison to polymeric ones. Availability of standard lithographic techniques and well-established technologies for silicon microfabrication are additional advantages of this approach, which opens up the route for a cost-efficient production in larger wafer sizes and higher volumes. Several Si-based technologies for the fabrication of metallic X-ray gratings were recently reported: conformal electroplating with partial (also known as spatial frequency doubling technique) [12] or complete [12,13,14] filling; seedless electroplating through a mask [15,16]; metal casting of Bi [17], as well as of Au [18] and Pb alloys [19]; atomic layer deposition (ALD) of Ir [20]; bottom-up Au electroplating on a metal seed layer [21,22,23,24,25]. Each of the above techniques has its own strengths and weaknesses. The spatial frequency doubling technique requires precise process control and suffers from a quality decrease when increasing the aspect ratio, especially in the lower pitch range. Metal casting is a very promising technique in terms of straightforward fabrication, but it cannot be performed with materials having a high X-ray absorbtion, such as pure Au, Pt, Ir, or Os due to their high melting temperature [19]. Gratings with an aspect ratio of 60:1 were demonstrated by ALD of Ir [20], but the method can be applied only to pitch sizes below 1 µm and small areas due to the very slow processing and the high cost of metal precursors. Seedless electroplating [15] is based on the oxidation of etched structures in low resistivity Si substrates, where the oxide is removed from the bottom of the trenches and the trenches are filled with Au by electroplating. This technique was recently used to fabricate structures with a pitch down to 1.3 µm and an Au height of 15 µm [16], which corresponds to an aspect ratio of 23:1. However, a higher aspect ratio for such small pitch was not feasible without lines distortions, while multistep processing raises the costs and lowers the yield of fabrication. The bottom-up Au electroplating in the presence of Bi additives is a powerful emerging maskless technique, which allows the filling of the recess structures with high aspect ratios. Gratings with a pitch size of 3 µm and an Au height of 85 µm (aspect ratio of 57:1) were recently demonstrated [23]. One drawback is that the method needs a conformal metal seeding, which for the highest aspect ratio structures requires a slow and expensive ALD coating. While not yet fundamentally limited by the process itself, an aspect ratio of 26:1 was reported for a pitch size of 1.3 µm [22]. Conformal Au electroplating or the damascene approach was demonstrated using ALD deposited Pt (pitch 200 nm, gold height 3.2 µm, aspect ratio 32:1) [13] and Ru (pitch 6 µm, gold height 180 µm, aspect ratio 60:1) [14] as plating seed layer. However, the ALD coating is slow and expensive, which sets severe limitations on the commercial production of gratings with an ALD processing step.

In this work, we used conformal electroplating of Au directly on low resistivity silicon wafers with a progressive closing of the recess structures and without any pre-deposition of a metallic seed layer, thanks to the high conductivity of the substrate. A similar approach was previously demonstrated for Ni electroplating of Vias structures in Si [26]. In this paper, we extended this process to Au electroplating and dense grating structures with much higher aspect ratios. We demonstrated the fabrication of metallic X-ray gratings on 4-inch wafer scale with pitch down to 1.2 µm and the height of the structures of 27 µm (aspect ratio up to 45:1). This makes them the smallest pitch gratings suitable for applications in X-ray energy range > 20 keV, required for GI in pathology, mammography, as well as in material science and failure analysis of weakly absorbing samples. Our future study includes the implementation of small pitch gratings in a commercial mammography system to increase the phase contrast sensitivity. The demonstrated 4-inch Si-based technology can be scaled up to larger wafer size and serial production. This is also particularly attractive for applications, such as mammography, since it enables the construction of interferometers with large field of view with a reduced number of gratings or even a single one, without the need of stitching multiple gratings [9].

## 2. Gratings Fabrication

The fabrication steps are schematically shown in Figure 1. Double side polished 250 µm thick low resistivity (<0.01 Ohm·cm), n-type (Sb doped) 4-inch silicon <100> wafers were used as substrates. The substrates were coated by a Cr 100 nm hard mask and photoresist layer. Either a displacement Talbot lithography [27,28] or a standard UV-lithography [16] step was performed for photoresist patterning. The underlying Cr mask was etched by Cl_2_/O_2_ plasma, the photoresist was stripped in acetone and the wafer rinsed with isopropanol and deionized (DI) water. The pattern consists of a periodic grating array with equal lines and spaces (duty cycle 0.5). We used a periodic transversal pattern consisting of sparse lines with the function of bridges to hold the thin silicon lamellas in the etched structures and to prevent lamella distortion during the electroplating process. In case of standard photolithography, the bridging structures were introduced directly into the mask design. This was not possible in a single step in the case of displacement Talbot lithography (DTL). Therefore, we introduced only the grating line structures using the process steps as described above. Then, the bridging structures were added in a second lithography step using a standard UV-lithography in positive photoresist. The bridge patterns were converted into Cr using a lift-off process. Deep reactive ion etching (DRIE) was performed in an Oxford Plasmalab100 tool using the Bosch process. Some details of the process are described in the next paragraph and in our recent work [16,29]. We typically etched structures to a depth of 25–35 µm, which is sufficient for X-ray interferometers working at energies up to about 30 keV. After removal of the Cr hard mask, an optional Au conductive layer was deposited onto the back side of the wafer to provide a uniform contact to the wafer during the electroplating process. After evaporation, the Au layer was annealed to ensure good adhesion and ohmic contact at the Au/Si interface. In order to ensure good surface conductivity, the sample was treated with vapor HF to remove the native oxide layer from the Si grating surface. This was an essential step to enable uniform electroplating on all surfaces. A short SF_6_ based plasma treatment has also shown good results but HF resulted in being more selective. Electroplating was performed in a cyanide based solution (Autronex-GVC from Enthone Inc.) using constant current in the range of 3.0–10 mA, corresponding to a current density in a range of ~1.5–5.0 µA/dm^2^ and a typical deposition time of about 2–4 days. The electrical contact of the wafer was done through the bottom of the wafer, ensuring a uniform growth of the initial layer throughout the complete wafer area [26]. A slight tapering of the Si structures produced during the DRIE step with a slow, constant current electroplating enabled a conformal deposition of Au and a complete filling of the trenches, as schematically shown in Figure 1, steps 6–8. 

The cross-sectional scanning electron microscope (SEM) images (Figure 2) illustrate a few examples of gratings etched by DRIE, while Table 1 summarizes the specific etching conditions. The time ratio between etching and deposition sequences during one Bosch etching loop was primarily adjusted for tuning the sidewall angle. This time-tuning alone was enough to obtain the targeted profiles up to a depth of ~30 µm in the case of a pitch *p* = 3.0 µm as in the presented example.

Higher aspect ratio etching is also possible by fine-tuning the RIE power, the ICP power, and the pressure conditions, as we recently reported elsewhere [29]. Gratings of various pitches in a range between 3.0 µm and 1.2 µm and a profile height of ~30 µm were etched on purpose with a slight tapered profile for favoring the conformal Au electroplating filling.

Figure 3 shows some examples of the produced Au gratings with conformal electroplating. It is noteworthy to mention that the Au grating with pitch of 1.2 µm and grating line height of 27 µm (see Figure 3d) corresponds to an aspect ratio of 45:1, which is almost twice that obtained by bottom-up Au electroplating approaches [16,22] of gratings in this pitch range. Further optimization of Si etching can also enable Au electroplating of gratings with higher aspect ratios. The described process requires fewer fabrication steps and is therefore much simpler compared to our previously reported method based on low resistivity wafers with the insulating mask and bottom-up electroplating [16]. Most remarkably, filling the Si structures with Au-electroplating does not require any metal seed layers—as reported in other approaches [12,13,14,21,22]—which substantially further simplifies the fabrication process. In particular, ALD is often used for the metal seeding layer in high aspect ratio structures, but it is a slow and expensive process. Our results indicate that the metallization by ALD coating prior to electroplating can be omitted by using low resistivity wafers with the presented method.

The minor drawback of the method is the formation of an extra Au layer on top of the gratings. With careful Au growth control the thickness of this extra layer can be kept as thin as a half width of the grating trench. In the case of small pitch gratings, the extra Au layer on top would cause a negligible absorption of the hard X-rays (>20 keV) in comparison to the absorption of the grating lines themselves. However, this drawback could become a limiting factor for gratings with large pitch (>10 µm). In this case, the gold layer formed on top of the lines and at the bottom of the grating trenches would act as an absorber, which (depending on the energy) may substantially reduce the X-ray flux. A further limitation is the need of a positive sidewall angle of the trench (see Figure 2b), which inevitably limits the maximum etching depth and directly affects the duty cycle of the grating. On the other hand, a vertical and, especially, a reversal tapered trench sidewall (see Figure 2a) would lead to an early closure of the trench during the electroplating step, and would introduce voiding defects. However, as is shown in Figure 3d, we did not observe the limitation of this effect for the gold electroplated structures with an aspect ratio, at least, up to 45:1. 

The presented method can also be applied for electroplating of other metals. For example, nickel is often used as a material for phase shifting X-ray grating fabrication, since it has much lower absorption than gold. The advantage of nickel compared to silicon is that it requires lower aspect ratio structures for target phase shifts, which might be preferred for applications with tight duty cycle tolerance requirements.

## 3. Proof of Concept and X-ray Application

The gratings were mounted on flat holders and tested in a Talbot–Lau interferometer consisting of an X-ray source, X-ray detector, two absorbing gratings (G0 and G2), a phase grating (G1), and the test sample to be imaged (see [2] for details). The pitch and the line height of the corresponding G0, G1, and G2 gratings are summarized in Table 2 along with the X-ray design energy, Talbot order, total interferometer length, and the measured mean visibility value. The visibility of the measured interference fringes, which are produced by the interferometer for a given Talbot order, is one of the most important parameters for grating characterization. The visibility reflects how strongly the fringes are modulated and is directly linked to the noise level in the retrieved images [30,31]. The absolute value of the visibility depends also on the X-ray spectrum, and the beam hardening caused by absorption in the substrates, and the eventual use of filters and threshold settings in the detector. In the ideal case of a grating with infinite absorbance of the grating lines, the visibility of the Talbot–Lau interferometer operated with a polychromatic X-ray source can exceed 30% at high Talbot orders [32]. In reality, state-of-the-art interferometers report values in a wide range around ~10–25%. Despite visibility being a characteristic of the whole interferometer (including the source and the detector), it is still considered to be a reliable figure of merit for the overall performance of the gratings. Two grating configurations were selected to demonstrate the performance of small pitch gratings in Talbot–Lau interferometers. Configuration 2 operates with a G2 that has half the pitch of G2 of Configuration 1, resulting in a halved total interferometer length.

The G1 and G2 gratings were designed to match the interferometric conditions for 5th Talbot order for both configurations. The G1 gratings were produced by Si DRIE, as schematically shown in steps 1 to 4 on Figure 1 [16,27]. They have a nominal duty cycle of 0.5 and were etched to obtain the height of Si structures corresponding to a phase shift of π- at the designed energy. The G2 gratings were etched in silicon and filled by Au conformal electroplating according to the method described above (steps 5–8 in Figure 1). Since the G0 gratings have a large pitch size, they were fabricated by using a gold filling method reported elsewhere [16]. 

Figure 4 reports on the X-ray characterization of the gratings. The interferometer was designed for 28 keV energy at the 5th Talbot order with a total interferometer length of 0.976 m. To collect the visibility map, we scanned the G2 grating over the interference pattern produced by the G1 grating. The visibility map reveals a good uniformity over the entire field of view of the setup (see Figure 4a). A sharp peak of the visibility histogram with a mean value at 16.5% indicates a highly uniform grating performance over the entire field of view. A differential phase contrast image was taken using a plastic capillary filled with glass colloidal particles as a test sample. The sample was placed in front of the G1 grating and shows a good quality of the image.

The interferometer of configuration 2 was designed for 26 keV energy at the 5th Talbot order and a total interferometer length of 0.556 m. The horizontal black line in the visibility map (see Figure 4b) is due to the dead region of the detector and has been excluded from the visibility histogram. The visibility strongly decays at the border regions of the field of view. This is due to the cone beam geometry of the X-ray source. The effect is more prominent in configuration 2, because of the almost half shorter interferometer length and almost twice higher aspect ratio of the G2 grating lines, both reducing the angular acceptance of the gratings. This issue can be addressed by bending the gratings to a curvature with bending radius equal to the source to grating distance [33]. While bending of the gratings was out of the scope of this paper, this is easily possible for a case of source to grating distance of ~0.5 m [9]. In order to extract the visibility without the influence of X-ray emission geometry, we selected only the central region of interest (ROI) of the field of view, as shown in the figure by the red dashed line, which properly reflects the performance of the gratings themselves. The mean visibility value in ROI is ~13%. This is slightly lower with respect to configuration 1 and can be caused both by a slight difference in the interferometric setup and alignment as well as by slight differences in the duty cycle and heights of the individual gratings in both interferometers. The measured visibility in configuration 2 indicates the remarkable quality of the interferometer with the small pitch grating (1.2 µm) and, at the same time, the increasing challenges for the characterization of small pitch gratings as the high aspect ratio increases (aspect ratio of 45:1).

## 4. Conclusions

A fabrication method to produce Au gratings in Si substrates was presented. The method is based on conformal electroplating of gold directly into structures etched in low resistivity Si wafers without the need of any metal seed deposition. The method requires only a few fabrication steps, making the processing more efficient compared to previous approaches. The silicon etching conditions were optimized to obtain slightly tapered sidewalls, which enabled gold electroplated filling of the trenches without voids. Gratings with pitches in the range 1.2–3.0 µm, heights in the range of 24–32 µm, and line aspect ratios up to 45:1 on a 4-inch wafer area were demonstrated. X-ray phase contrast imaging experiments reveal good performance using a conventional X-ray tube in Talbot–Lau configuration. Decreasing the pitch increases the aspect ratio of the grating lines and decreases the length of the interferometer, both calling for the necessity to bend the gratings to address the divergence of the beams of the laboratory X-ray sources and ensure visibility uniformity on a large field of view.

We also see a high potential to combine the presented method with the bottom-up Au electroplating technique for efficient grating fabrication [21,22,23] or with the partial conformal coating method [12], as the metallization step can be avoided, in order to further simplify the process and reduce the fabrication costs.

## Figures and Tables

**Figure 1 micromachines-12-00517-f001:**
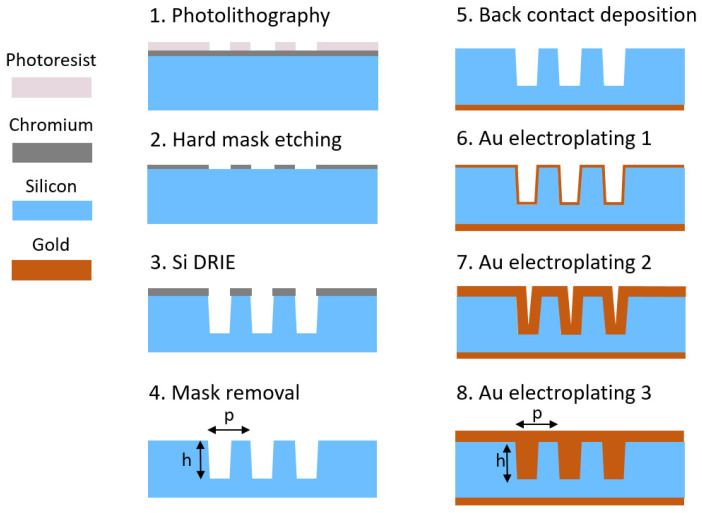
Fabrication steps. Steps 1–4: fabrication of high aspect ratio Si structures with a pitch *p* and a height h and a tapered profile. Step 5: deposition of the ohmic Au contact on the back side of the wafer. Step 6: seedless formation of the initial Au layer directly on the surface of the grating. Step 7: intermediate stage of Si trench filling by Au. Step 8: complete filling of Si trenches with Au.

**Figure 2 micromachines-12-00517-f002:**
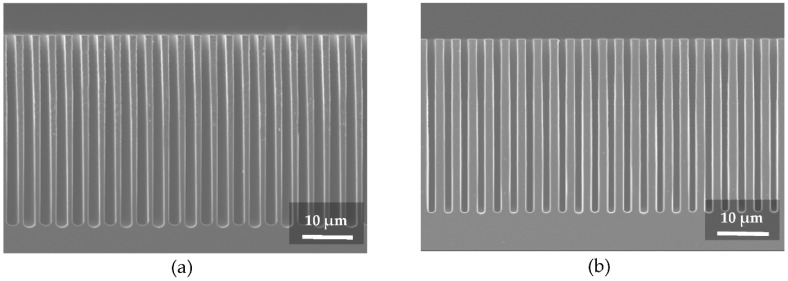
Example of sidewall shape tuning in Si grating with a pitch of 3 µm. Changing the ratio between the times of etching and deposition steps (as shown in Table 1a,b) results in: (**a**) etching with a negative sidewall angle; (**b**) etching with a slight positive sidewall angle, which is suitable for conformal electroplating.

**Figure 3 micromachines-12-00517-f003:**
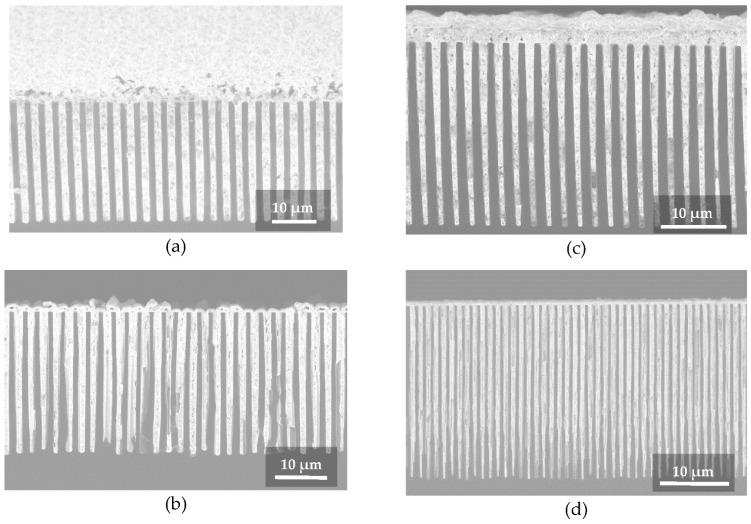
Cross-section SEM images of Au/Si gratings fabricated by seedless conformal electroplating: (**a**) *p* = 3.0 µm, h = 31 µm (image is taken at 30° tilt angle); (**b**) *p* = 2.68 µm, h = 30 µm; (**c**) *p* = 2.4 µm, h = 29 µm; (**d**) *p* = 1.2 µm, h = 27 µm.

**Figure 4 micromachines-12-00517-f004:**
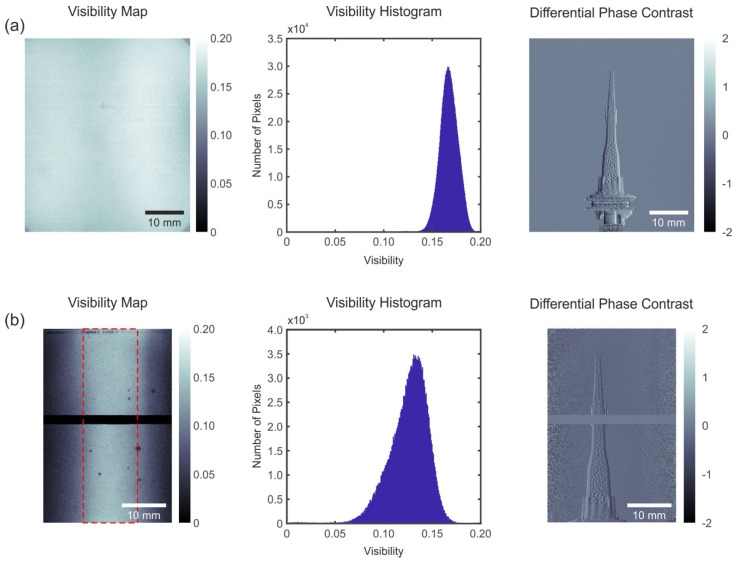
X-ray grating test results. Visibility map, visibility histogram and differential phase contrast image for two configurations shown in Table 2: (**a**) configuration 1; (**b**) configuration 2. The visibility histogram in case (**a**) was measured through the whole area, while in case (**b**) only in the central part outlined by the dash line. The visibility reduction in the side areas of case (**b**) is due to a twice higher aspect ratio of the G2 grating lines and almost a twice shorter interferometer length, which strongly limits the acceptance angle of the flat gratings. Details on the measurements can be found elsewhere [2].

**Table 1 micromachines-12-00517-t001:** Etching parameters of grating shown in Figure 2a,b, correspondingly.

**(a)** **Step:**	**Temperature [°C]**	**Pressure [mTorr]**	**RF Power [W]**	**ICP Power [W]**	**SF6 [sccm]**	**C4F8 [sccm]**	**Time [sec]**	**Loop Count**
Etching	0	20	30	600	100	5	4	350
Deposition	0	20	20	600	5	100	3	
**(b)** **Step:**	**Temperature [°C]**	**Pressure [mTorr]**	**RF Power [W]**	**ICP Power [W]**	**SF6 [sccm]**	**C4F8 [sccm]**	**Time [sec]**	**Loop Count**
Etching	0	20	30	600	100	5	3	400
Deposition	0	20	20	600	5	100	3	

**Table 2 micromachines-12-00517-t002:** Different tested grating configurations. (Fabrication of G0 gratings is described in [16]. Details on the measurement setup are presented in).

Configu-ration	G0 Pitch/Height, µm	G1 Pitch/Height, µm	G2 Pitch/Height, µm	Design Energy, keV	Talbot Order	Interferometer Length, m	Measured Mean Visibility, %
1	7.2/38	3.6/36	2.4/30	28	5	0.976	16.5
2	8.9/40	2.1/33	1.2/27	26	5	0.556	13

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
