# Peer review of "Fabrication of X-ray Gratings for Interferometric Imaging by Conformal Seedless Gold Electroplating"

_micromachines, 2021, doi:10.3390/mi12050517_

Round 1

Reviewer 1 Report

The paper "Fabrication of X-ray gratings for interferometric imaging by conformal seedless gold electroplating" presents a method to produce "small" pitch gratings with high aspect ratios, used for X-ray interferometric imaging. Using a combination of deep reactive ion etching and Au electroplating, they are able to fabricate such devices, with void-free Au filling off the trenches. Gratings are then tested and shown to perform well.

The paper is sound and describes the methodology in details. It contains sufficient illustrations. This paper is suitable for publication.

Author Response

We thank the reviewer for the positive feedback.

Reviewer 2 Report

Dear authors,

Please consider the suggested comments to improve the quality of the current version of the manuscript:

  1. In the abstract, please include the details of the limitations of the current/previous technologies and specify the scientific advances made in the proposed fabrication technique to produce small pitch gratings for X-ray interferometric imaging applications, to overcome those limitations.
  2. In the current version of the abstract, only the summary of the work is mentioned in a very broad view. Instead please include specific details of the research work presented in the manuscript. The abstract needs to be incorporated with the gist of the complete work in the manuscript.
  3. Please update the abstract with the experimental conditions, parameters considered, and scientific advancements made, as the abstract of a scientific research paper should be precisely mentioning the specific research question that is answered, experimental conditions, operational parameters, results, and conclusions. The current version of the abstract is more as an introduction to the study & research performed rather than explaining the actual scientific advancements made.
  4. In the introduction, please include the knowledge gaps existing in the current research work and prior studies performed in the field. Very importantly, please specify the need for the current work presented in the manuscript.
  5. In the last paragraph of the introduction, kindly include the details of the broader impacts on the study made and the results achieved. It is very important to provide the future scope of the research performed to make a strong impact on the readers on the research performed/Study proposed.
  6. The specification of the ‘lateral etching’ and the corresponding preventive measures was not mentioned in the manuscript. As the ‘lateral etching’ effect is very important to be mentioned while discussing the bosch etching technique, please include the specifications and details of the lateral etching caused.
  7. Please include the AFM images of the etched surface to discuss the surface finish achieved in the process.
  8. In the results section, the discussion on the surface morphology is one of the important factors that need to be addressed especially while proposing the new micro/nano fabrication techniques incorporated with the etching process.
  9. The section-4 is missing in the manuscript. Please arrange the sections in the manuscript as per the formatting guidelines of the journal. Kindly refer to the author's instructions to understand the formatting process during the submission of the manuscript. Please use the link below for author instructions: https://www.mdpi.com/journal/micromachines/instructions
  10. Please revise the manuscript with English grammar. There are a few places that the manuscript needs to be improved with respect to English writing.

Round 2

Reviewer 2 Report

Dear authors,
Thank you for updating the manuscript with recommended changes.